# Relationship between the Central and Peripheral Thyroid Sensitivity Indices and Fetal Macrosomia: A Cohort Study of Euthyroid Pregnant Women in China

**DOI:** 10.3390/diagnostics13122013

**Published:** 2023-06-09

**Authors:** Xin Zhao, Jianbin Sun, Ning Yuan, Xiaomei Zhang

**Affiliations:** Endocrinology Department, Peking University International Hospital, Beijing 100001, China; zhaoxin1@pkuih.edu.cn (X.Z.);

**Keywords:** thyroid sensitivity, gestational diabetes mellitus, macrosomia, glycosylated hemoglobin

## Abstract

(1) Background: To explore the correlation between central and peripheral thyroid sensitivity indices and macrosomia in euthyroid pregnant women and to provide clinical basis for the prevention and treatment of macrosomia. (2) Methods: This study is a prospective study. A total of 1176 euthyroid women in early pregnancy in the obstetrics department of Peking University International Hospital from December 2017 to March 2019 were enrolled. The women were divided into two groups, namely the macrosomia and non-macrosomia groups, according to birth weight. (3) Results: The level of free triiodothyronine (FT3), thyroid-stimulating hormone (TSH), thyroid feedback quantile-based index (TFQI), thyrotropin-T4 resistance index (TT4RI), thyroid-stimulating hormone index (TSHI), and free triiodothyronine/free thyroxine (FT3/FT4) in the macrosomia group was higher than that in the non-macrosomia group (*p* < 0.05). The multivariate logistic regression model showed that FT3, TFQI, TT4RI, TSHI, and FT3/FT4 were independent risk factors for macrosomia in early pregnancy after adjusting for age, body mass index, parity, blood pressure, blood glucose, and blood lipid levels (*p* < 0.05, respectively). (4) Conclusions: TFQI, TT4RI, TSHI, and FT3/FT4 are independent risk factors for fetal macrosomia in early pregnancy in euthyroid women.

## 1. Introduction

With the growth of the economic level and the change of human diet structure, the risk of many endocrine diseases during pregnancy has significantly increased [1,2], including gestational diabetes mellitus (GDM), thyroid-related diseases, and hyperlipidemia. The role of thyroid hormone (TH) runs through the whole pregnancy process. It changes with the physiological changes of pregnancy. During pregnancy, in order to meet the additional hormone demand, the thyroid volume will increase by about 40%, accompanied by the increase in triiodothyronine (T3) and thyroxine (T4) secretion and metabolism at different levels. Macrosomia refers to the newborn whose birth weight exceeds 4000 g [3]. Macrosomia can significantly increase the risk of complications during maternal and fetal delivery [4]. Macrosomia is an obstetric complication that affects 10% of all pregnancies and is associated with severe maternal–fetal complications such as maternal birth canal trauma, fracture of the clavicle, brachial plexus injury, and perinatal asphyxia. It is associated with increased risks of cesarean section and trauma to the birth canal and fetus [3]. One of the main concerns in pregnancies with macrosomia is that macrosomic fetuses show different cardiovascular indices, in particular the umbilical artery pulsatility index, compared with non-macrosomic fetuses [5]. Early identification of risk factors can allow preventive measures to be taken to avoid adverse perinatal outcomes.

At present, it is generally believed that maternal clinical hypothyroidism during pregnancy is related to fetal intrauterine growth restriction and birth weight loss [6], but recent studies have reported that maternal hypothyroidism will also increase the risk of macrosomia [7]. However, for euthyroid pregnant women, there are few research reports on the relationship between TH and neonatal weight, and the research conclusions are inconsistent [8,9].

Impaired TH sensitivity can be divided into central sensitivity impairment and peripheral sensitivity impairment. Central sensitivity impairment is manifested in the influence of the feedback circuit of the hypothalamus–pituitary–thyroid (HPT) axis of the central system, such as the appearance of without thyroid disease syndrome under long-term fasting or certain disease conditions (such as morbid obesity) [10,11]; the impairment of peripheral sensitivity is manifested by the reduction in hormone metabolism. At present, the commonly used indices to reflect TH sensitivity include the thyrotropin-T4 resistance index (TT4RI), thyroid-stimulating hormone index (TSHI), thyroid feedback quantile-based index (TFQI), and free triiodothyronine/free thyroxine (FT3/FT4). TSHI, TT4RI, and TFQI reflect the change of TH sensitivity to the HPT axis [12,13], and FT3/FT4 reflects the sensitivity of the peripheral tissue to FT4 changes.

This study aims to analyze the correlation between TH, central and peripheral thyroid hormone sensitivity, and macrosomia in euthyroid pregnant women, and establish a prediction model for macrosomia by using relevant indices so as to provide new clinical evidence-based medical evidence for the prevention and treatment of macrosomia in clinical practice.

## 2. Materials and Methods

### 2.1. Research Subjects

This study is a prospective study. A total of 1176 euthyroid women in early pregnancy in the obstetrics department of Peking University International Hospital from December 2017 to March 2019 were enrolled. All pregnant women were included in the study for 7–12 weeks. The pregnant women and the birth outcome of the fetus were regularly followed up, and the birth height and weight of the fetus were recorded.

The following are the inclusion criteria: (1) at least 18 years old; (2) thyroid function is normal in early pregnancy; and (3) willing to enter the queue and accept the survey of relevant questionnaires and agree to collect blood samples after informing the relevant survey content.

The exclusion criteria are as follows: (1) pregnant women with cardiovascular and cerebrovascular diseases, respiratory diseases, pre-pregnancy diabetes, thyroid diseases, blood system diseases, and liver and kidney diseases diagnosed in early pregnancy; (2) women with multiple pregnancies; and (3) missing basic data.

This study was approved by the Hospital Ethics Review Committee. All respondents have been informed of the purpose of this investigation and signed the informed consent form when entering the study.

Fetal macrosomia was defined as a live birth of more than 4000 g.

### 2.2. Research Methods

#### 2.2.1. General Situation

The age, parity, and gestational week of pregnant women were recorded at the time of enrollment; blood pressure (including systolic blood pressure and diastolic blood pressure), height, and weight were collected and measured; and the body mass index was calculated and recorded. The BMI was calculated using the following formula: BMI (kg/m^2^) = weight (kg)/body height^2^ (m^2^).

#### 2.2.2. Biochemical Indices

All the subjects had fasting 5 mL venous blood collected in the morning during 7–12 weeks of gestation. The biochemical indices include glycosylated hemoglobin (HbA1c), fasting blood glucose (FBG), postprandial blood glucose (PBG), triglyceride (TG), total cholesterol (TC), high-density lipoprotein cholesterol (HDL-C), low-density lipoprotein cholesterol (LDL-C), serum creatinine (sCr), uric acid (UA), homocysteine (Hcy), FT4 (free thymidine), FT3 (free triiodothyronine), and TSH (thyroid-stimulating hormone). HbA1c is determined by high-performance liquid chromatography.

#### 2.2.3. Thyroid Hormone Sensitivity Assessment Index


(1)TFQI: TFQI = cdfFT4 − (1-cdfTSH). First, the FT4 and TSH of pregnant women in this clinical study were sorted from the minimum value to the maximum value. Then, according to the principle of the empirical cumulative distribution function (CDF), the probability distribution of FT4 and TSH in the population is converted to the probability quantile between 0 (representing the percentage of people below this value in the overall population is 0) and 1 (representing the percentage of people below this value in the overall population is 100%). After formula calculation, the value range of the TFQI is −1 to 1. The TFQI is negative, indicating that the HPT axis is relatively more sensitive to FT4 changes. A positive value indicates that the HPT axis is relatively insensitive to FT4 changes; 0 indicates that the sensitivity of the HPT axis to the FT4 change is normal.



(2)TT4RI and TSHI


TT4RI = FT4 × TSH

TSHI = LnTSH + 0.1345 × FT4

The higher the TSHI and TT4RI values, the lower the sensitivity of the center to thyroid hormones.


(3)FT3/FT4 index


FT3/FT4 = FT3/FT4

The higher the FT3/FT4 value, the higher the sensitivity of the peripheral to thyroid hormone.

#### 2.2.4. Definitions of GDM and HDP

The pregnant women were screened for gestational diabetes mellitus by 75 g oral glucose tolerance test at 24–28 weeks of gestation.

IADPSG is used as the diagnostic criteria for gestational diabetes [14], that is, 75 g oral glucose tolerance test, fasting blood glucose, and blood glucose 1 h and 2 h after taking glucose. The three blood glucose values should be lower than 5.1 mmol/L, 10.0 mmol/L, and 8.5 mmol/L, respectively. If any blood glucose value reaches or exceeds the above criteria, GDM is diagnosed.

HDP included gestational hypertension, preeclampsia–eclampsia, chronic hypertension (of any cause diagnosed before 20 weeks of gestation), and chronic hypertension with preeclampsia superimposed [15].

### 2.3. Statistical Methods

All data were analyzed using SPSS 22.0. Data were tested for normality, and normally distributed data were expressed as means ± standard deviation (x ± s) and compared using *t*-tests, while non-normally distributed data were expressed as medians (P25, p75) and compared using rank-sum tests. The statistical description of counting data is based on the constituent ratio or rate, and comparisons between the two groups were made using χ^2^ tests. An unconditional logistic regression model was used for single-factor and multifactor analyses of related factors to calculate the OR and 95% CI. The receiver operating characteristic (ROC) curves were plotted, and the area under the ROC curve (AUC) was calculated. All statistical tests were two-sided, and *p* < 0.05 was considered statistically significant.

## 3. Results

### 3.1. Comparison of General Conditions and Biochemical Indices between Two Groups in the First Trimester

Among the 1176 pregnant women, 107 mothers gave birth to a newborn with weight of more than 4000 g. The rate of macrosomia was 9.10%. The BMI of the macrosomia group was higher than that of the non-macrosomia group (22.05 ± 3.28 vs. 22.14 ± 2.91, t = −1.95, *p* < 0.05). The levels of HbA1c and FBG in the macrosomia group were significantly higher than those in the non-macrosomia group in the first trimester of pregnancy (5.10 ± 0.44 vs. 5.24 ± 0.28, t = −2.54, *p* < 0.05; and 4.61 ± 0.52 vs. 5.04 ± 0.38, t = −2.94, *p* < 0.05, respectively). In addition, compared with the two groups, TC and TG in the macrosomia group were significantly higher in the first trimester of pregnancy than those in the non-macrosomia group (3.95 ± 0.69 vs. 4.03 ± 0.96, t = −2.40, *p* < 0.05; and 1.01 ± 0.63 vs. 1.18 ± 0.63, t = −2.91, *p* < 0.05, respectively). SBP in the macrosomia group was higher than that in the non-macrosomia group in the first trimester of pregnancy (109.85 ± 10.89 vs. 112.13 ± 12.60, t = −2.22, *p* < 0.05), but there was no significant difference in DBP between the two groups (66.26 ± 10.09 vs. 67.94 ± 10.02, t = −1.48, *p* = 0.14). The levels of FT3 and FT3/FT4 in the macrosomia group were significantly higher than those in the non-macrosomia group (4.62 ± 0.50 vs. 4.96 ± 0.50, t = −6.74, *p* < 0.05; and 0.27 ± 0.04 vs. 0.29 ± 0.03, t = −2.27, *p* < 0.05, respectively). The level of FT4 in the macrosomia group was significantly lower than that in the non-macrosomia group (17.51 ± 1.47 vs. 16.79 ± 1.88, t = 3.69, *p* < 0.05). In early pregnancy, the proportion of GDM in the macrosomia group was significantly higher than that in the non-macrosomia group (18.43% vs. 26.19%, χ^2^ = 4.17, *p* < 0.05). The levels of TFQI, TT4QI, and TSHI in the macrosomia group were significantly higher than those in the non-macrosomia group (−0.02 ± 0.47 vs. 0.22 ± 0.20, t = −4.98, *p* < 0.05; 25.99 ± 6.31 vs. 32.29 ± 5.39, t = −4.29, *p* < 0.05; and 2.52 ± 0.65 vs. 2.66 ± 0.63, t = −2.27, *p* < 0.05, respectively). There was no significant difference in age, UA, sCr, Hcy, and HDP ratio between the two groups (30.88 ± 3.76 vs. 31.12 ± 3.80, t = 0.26, *p* = 0.79; 216.48 ± 48.22 vs. 220.79 ± 43.70, t = −1.12, *p* = 0.26; 49.48 ± 6.81 vs. 49.65 ± 7.89, t = −0.25, *p* = 0.81; 6.57 ± 1.55 vs. 6.69 ± 2.39, t = −0.69, *p* = 0.49; and 2.43% vs. 2.80%, χ^2^ = 0.06, *p* = 0.8, respectively) (Table 1 for details).

### 3.2. Logistic Regression Analysis of TH and Sensitivity Indices with Macrosomia

The multivariate logistic regression model was established with macrosomia as the dependent variable and the indices that are statistically significant based on univariate analysis as the independent variables. The results showed that after adjusting for age, BMI, parity, blood pressure, blood lipid, and blood glucose levels, the high level of FT3/FT4 was an independent risk factor for macrosomia in early pregnancy (aOR = 3.14, 95% CI 1.14, 8.67), and TT4RI and TSHI are independent risk factors for macrosomia (aOR = 3.38, 95% CI 1.23, 9.31; aOR = 3.66, 95% CI 1.14, 11.77; aOR = 2.97, 95% CI 1.09, 8.13) (Table 2 for details).

### 3.3. Univariate Prediction Model of Macrosomia

According to the model of single-variable FT3, FT4, FT3/FT4, TFQI, TT4RI, and TSHI, to predict the risk of macrosomia, the area under the ROC curve (AUC) is, from large to small, FT3 > TFQI > TSHI > TT4RI > FT4 > TSHI > FT3/FT4, and the corresponding points are 4.95 pmol/L, 0.73, 3.57, 30.70, 19.15 pmol/L, 3.57, and 0.27, respectively (Table 3 and Figure 1 for details).

### 3.4. Multivariable Prediction Model of Macrosomia Risk

The multivariable prediction model was established with macrosomia as the dependent variable and parity, blood pressure, blood lipid, blood glucose, FT4/FT3, TFQI, TT4RI, and TSHI as independent variables. The area under the ROC curve was 0.77 (95% CI 0.69, 0.85), the specificity was 67.20%, the sensitivity was 75.86%, and the accuracy was 67.83% (Figure 2 for details).

## 4. Discussion

Thyroid is an important endocrine organ of the human body. TH participates in human energy metabolism and nervous system development and plays an important role in maintaining the normal growth of the body [16]. During pregnancy, normal thyroid function plays an important role in maintaining the physiological balance of pregnant women and the normal growth and development of the fetus. TH during pregnancy affects the normal development of the placenta; controls fetal growth by promoting placental formation; regulates metabolism, fetal glucose, oxygen consumption, and other auxiliary factors; and directly affects bone growth, tissue differentiation, and proliferation. During pregnancy, the endocrine status of pregnant women has greatly changed, so the thyroid function has also undergone a series of corresponding changes, mainly including four aspects: the TSH-like effect of HCG, the increase in the number of thyroxine binding globulin, the increase in the content of type III placental deiodinase, and the increase in the iodine clearance rate of the kidney.

During pregnancy, the maternal thyroid hormone reaches the fetus through the placenta. TH may mediate the lipid and protein metabolism of the fetus. Low TH may cause the metabolism of the fetus to slow down during the growth process and increase the birth weight. In recent years, there are many studies on the correlation between thyroid disorders in pregnancy and perinatal adverse maternal and infant outcomes. Most of the reports are about the relationship between thyroid function in pregnancy and neonatal weight. At present, many research results support that hypothyroidism in pregnancy will increase the risk of macrosomia. FT4 in the normal range of pregnant women is negatively related to the weight of the newborn, and the relationship between FT4 and birth weight is more obvious in the middle and late pregnancy than in the early pregnancy [17]. A recent meta-study shows that [18], compared with euthyroid precursors, isolated hyperoxinamia (IH) precursors were associated with an increased risk of macrosomia (RR 1.62 [CI 1.31–2.02]; I^2^ = 42%).

The results of this study show that compared with the non-macrosomia group, the FT4 level of the macrosomia group in early pregnancy is significantly lower, which is consistent with the results of previous studies. A recent meta-analysis included nine prospective cohort studies and one case control study, and the results found that the maternal FT4 and TSH levels during pregnancy are negatively correlated with the birth weight of the newborn [19]. When FT4 is relatively sufficient, it directly promotes the intrauterine growth of the fetus and also reduces the resistance of peripheral blood vessels, especially placental blood vessels, thus increasing the nutrient transport capacity of the placenta [20]. Previous studies have suggested that simple hypothyroidism in early pregnancy is related to premature delivery and macrosomia, as well as weight gain of and head circumference increase in newborns [21]. However, in further single-factor regression analysis, this study did not find that a low FT4 level is a risk factor for macrosomia. The analysis may be because the subjects in this study are euthyroid pregnant women, and the FT4 level is within the normal range, which cannot reflect the correlation between low FT4 blood syndrome and macrosomia.

The negative feedback regulation of the pituitary–thyroid axis is an important mechanism to maintain the stability of TH. However, FT3 or TSH alone may not be adequate to reflect the regulation of TH homeostasis. Therefore, the sensitivity to TH, including TT4RI, TSHI, and TFQI, was proposed as a quantitative marker of pituitary–thyroid-stimulating function to comprehensively interpret the thyroid status. TT4RI is a simple index for evaluating the pituitary sensitivity of thyroid hormone proposed by Jostel et al. for the first time [22]. It is used to estimate the maximum pituitary TSH reserve by extrapolating the TSH feedback inhibition amount to the standardized unsuppressed TSH with the FT4 value. It quantifies the deviation of the response of the middle pituitary gland to TH in a continuous way, that is, the deviation between the measured value of TSH and the actual value. Comprehensive indicators TSHI, TT4RI, and TFQI were recommended as new central TH resistance indicators. The advantages are that they will not produce an extremely high value in the case of abnormal thyroid function, and they are relatively stable. Recent studies have shown that a higher thyroid hormone sensitivity index is associated with obesity, metabolic syndrome, diabetes, and diabetes-related mortality [23,24,25]. In a representative sample of the Iranian population, thyroid hormone resistance as presented by the newer TFQI was significantly associated with DM and high BP in euthyroid subjects.

In this study, we first explored the relationship between thyroid hormone central sensitivity and macrosomia in euthyroid pregnant women. Previous studies have confirmed that gestational diabetes mellitus (GDM) is a risk factor for macrosomia [26,27,28]. It is reported that 15–45% of GDM pregnant women may give birth to giant babies, which is three times higher than that of non-GDM pregnant women. The results of this study also confirmed that the proportion of GDM in the macrosomia group was significantly higher than that in the non-macrosomia group (*p* < 0.05). When this study further explored the relationship between thyroid hormone sensitivity and macrosomia, GDM may bias the results. Therefore, when we further studied the relationship between thyroid hormone sensitivity and macrosomia, we adjusted the occurrence of GDM. The results showed that the indices of central thyroid sensitivity were still independent risk factors for macrosomia. The relationship and mechanism between the thyroid hormone axis in pregnancy and neonatal birth weight still need further study.

The manifestations of resistance to thyroid hormone (RTH) syndrome vary as a spectrum of clinical findings between hypothyroidism and hyperthyroidism in different tissues, suggesting that due to the organ’s main receptors, which can be TRβ (liver, adipose tissues) or TRα (skeletal and cardiac muscles, vascular endothelium), TH deficiency, sufficiency, and excess may coexist at the same time in a patient. There are three types of deiodinase (DIO) in peripheral tissues that regulate thyroid hormone metabolism, namely type 1, type 2, and type 3 iodothyronine deiodinase. T4 is usually considered as a pre-hormone. As the substrate of bioactive form T3, the ratio of FT3/FT4 is used to indicate deiodinase activity. A cross-sectional study showed that a higher BMI was associated with increased deiodinase activity in women with normal TH, and higher deiodinase activity was significantly associated with a higher glucose level. FT3 is mainly converted from iodothyronine DIO in permanent issues, and FT3/FT4 can reflect the permanent sensitivity, that is, the sensitivity of permanent organizations and issues to the change of FT4. The results of this study show that the levels of FT3 and FT3/FT4 in the first trimester of pregnancy in the macrosomia group are higher than those in the non-macrosomia group, and FT3/FT4 is an independent risk factor for the occurrence of macrosomia. This is the first time to analyze the relationship between peripheral thyroid sensitivity and macrosomia in pregnant women.

In this study, we used the ROC curve to evaluate the predictive value of FT3, FT4, FT3/FT4, TFQI, TT4RI, and TSHI for macrosomia. The area under the ROC curve ranked from large to small is as follows: FT3 > TFQI > TSHI > TT4RI > FT4 > TSHI > FT3/FT4. At the same time, the results of this study show that parity, HDP, blood lipids, and GDM are independent risk factors for macrosomia, and these results have been fully confirmed in early studies [29,30,31]. In this study, we used macrosomia as the dependent variable and took parity, blood pressure, blood lipids, GDM, FT4/FT3, TFQI, TT4RI, and TSHI as the independent variables to establish a multivariable prediction model. The area under the ROC curve of the prediction model was 0.77 (95% CI 0.69, 0.85), the specificity was 67.20%, the sensitivity was 75.86%, and the accuracy was 67.83%, which had a good predictive value.

This study explored the relationship between peripheral and central thyroid sensitivity and macrosomia and excluded the effects of thyroid dysfunction, autoimmune diseases, and taking of thyroid hormone drugs so as to minimize the influencing factors. However, there are still some limitations in this study. First of all, this study is a single-center hospital study with a limited sample size, so the research results have certain limitations. In future studies, we should use multicenter hospitals and large-sample patients to conduct research, eliminate factors that affect the results, and further verify the results of this study. Second, our study only included thyroid function indicators in the early stage of pregnancy and failed to dynamically assess the thyroid function level during the complete pregnancy. In future studies, thyroid function levels in different stages of pregnancy can be included to better reveal the relationship between thyroid sensitivity and macrosomia. In addition, in future studies, we will further evaluate the research on the mechanism between thyroid sensitivity and macrosomia and clarify the correlation between the thyroid hormone axis in pregnancy and newborn birth weight.

## 5. Conclusions

By analyzing the correlation between the thyroid sensitivity of euthyroid pregnant women in early pregnancy and macrosomia, this study found that the central and peripheral thyroid sensitivities in early pregnancy were independent risk factors for macrosomia and established a prediction model for macrosomia using relevant indicators, which provided a theoretical basis for the prevention and treatment of macrosomia clinically.

## Figures and Tables

**Figure 1 diagnostics-13-02013-f001:**
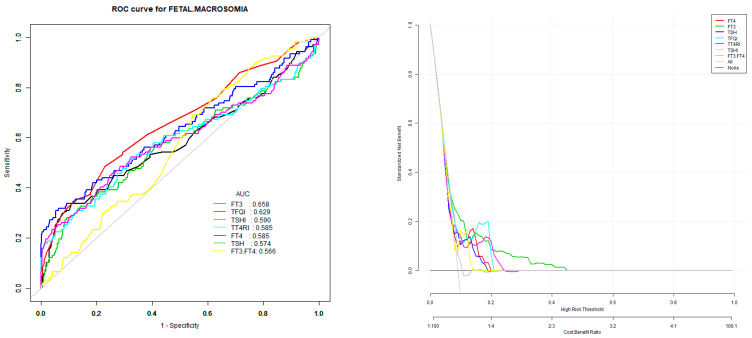
ROC curves for the accuracy of macrosomia in patients. Note: FT4 is for free thyroxine, FT3 is for free triiodothyronine, TSH is for thyroid-stimulating hormone, TT4RI is for thyrotroph thyroxine resistance index, TSHI is for thyroid-stimulating hormone index, and TFQI is for thyroid feedback quantile-based index.

**Figure 2 diagnostics-13-02013-f002:**
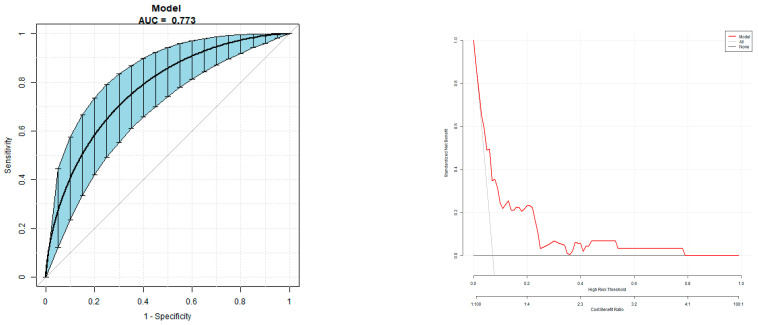
Overall predictive accuracy of multivariate predictive model for the risk of fetal macrosomia. AUC is 0.77 (95%CI 0.69, 0.85), with specificity of 67.20%, sensitivity of 75.86%, and accuracy of 67.83%.

**Table 1 diagnostics-13-02013-t001:** Comparison of general conditions and biochemical indices between two groups.

Index	Non-Macrosomia Group (n = 1069)	Macrosomia Group (n = 107)	t (χ^2^)	*p*
Age (years)	30.88 ± 3.76	31.12 ± 3.80	0.26	0.79
BMI (kg/m^2^)	22.05 ± 3.28	22.14 ± 2.91	−1.95	<0.05
Parity				
0	635 (59.40%)	52 (48.60%)		
≥1	434 (40.60%)	55 (51.40%)	9.94	<0.05
SBP (mmHg)	109.85 ± 10.89	112.13 ± 12.60	−2.22	<0.05
DBP (mmHg)	66.26 ± 10.09	67.94 ± 10.02	−1.48	0.14
TC (mmol/L)	3.95 ± 0.69	4.03 ± 0.96	−2.40	<0.05
TG (mmol/L)	1.01 ± 0.63	1.18 ± 0.63	−2.91	<0.05
LDL-C (mmol/L)	2.11 ± 0.55	2.12 ± 0.54	−0.31	0.76
HDL-C (mmol/L)	1.42 ± 0.29	1.40 ± 0.27	0.68	0.50
UA (umol/L)	216.48 ± 48.22	220.79 ± 43.70	−1.12	0.26
sCr (umol/L)	49.48 ± 6.81	49.65 ± 7.89	−0.25	0.81
HbA1c (%)	5.10 ± 0.44	5.24 ± 0.28	−2.54	<0.05
FBG (mmol/L)	4.61 ± 0.52	5.04 ± 0.38	−2.94	<0.05
Hcy (umol/L)	6.57 ± 1.55	6.69 ± 2.39	−0.69	0.49
FT4 (pmol/L)	17.51 ± 1.47	16.79 ± 1.88	3.69	<0.05
FT3 (pmol/L)	4.62 ± 0.50	4.96 ± 0.50	−6.74	<0.05
TSH (uIU/mL)	1.58 ± 0.35	1.89 ± 0.36	−3.39	<0.05
TFQI	−0.02 ± 0.47	0.22 ± 0.20	−4.98	<0.05
TT4RI	25.99 ± 6.31	32.29 ± 5.39	−4.29	<0.05
TSHI	2.52 ± 0.65	2.66 ± 0.63	−2.04	<0.05
FT3/FT4	0.27 ± 0.04	0.29 ± 0.03	−2.27	<0.05
GDM (%)	197 (18.43%)	28 (26.19%)	4.17	<0.05
HDP (%)	26 (2.43%)	3 (2.80%)	0.06	0.81

Note: BMI is for body mass index, SBP is systolic blood pressure, DBP is for diastolic blood pressure, FBG is for fasting blood glucose, HbA1c is for glycosylated hemoglobin, sCr is for serum creatinine, Hcy is for homocysteine, UA is for uric acid, TC is for total cholesterol, TG is for triglycerides, LDL-C is for low-density lipoprotein cholesterol, HDL-C is for high-density lipoprotein cholesterol, FT4 is for free thyroxine, FT3 is for free triiodothyronine, TSH is for thyroid-stimulating hormone, TT4RI is for thyrotroph thyroxine resistance index, TSHI is for thyroid-stimulating hormone index, TFQI is for thyroid feedback quantile-based index, GDM is for gestational diabetes mellitus, and HDP is for hypertensive disorders of pregnancy.

**Table 2 diagnostics-13-02013-t002:** Logistic regression analysis of TH and sensitivity indices with macrosomia.

Index	Crude OR	95% CI	*p*	Adjusted OR (aOR)	95% CI	*p*
Thyroid hormone						
FT3 (pmol/L)						
Low	1			1		
Medium	0.98	0.55, 1.75	0.94	1.29	0.47, 3.56	0.62
High	2.80	1.73, 4.56	<0.05	3.15	1.25, 7.97	<0.05
FT4 (pmol/L)						
Low	1			1		
Medium	0.68	0.40, 1.18	0.17	0.45	0.16, 1.31	0.14
High	0.61	0.63, 1.62	0.76	0.82	0.50, 3.00	0.66
TSH (uIU/mL)						
Low	1			1		
Medium	0.89	0.52, 1.53	0.68	0.84	0.30, 2.31	0.73
High	1.75	1.09, 2.82	<0.05	1.29	0.51, 3.25	0.60
Peripheral thyroid resistance index						
FT3/FT4						
Low	1			1		
Medium	2.80	1.59, 4.91	<0.05	1.31	0.48, 3.57	0.60
High	2.57	1.45, 4.54	<0.05	3.14	1.14, 8.67	<0.05
Central thyroid resistance index						
TFQI						
Low	1			1		
Medium	3.50	1.71, 7.17	<0.05	1.58	0.53, 4.64	0.41
High	6.68	3.37, 13.23	<0.05	3.38	1.23, 9.31	<0.05
TT4RI						
Low	1			1		
Medium	2.43	1.29, 4.60	<0.05	2.91	0.88, 9.64	0.08
High	4.40	2.41, 8.03	<0.05	3.66	1.14, 11.77	<0.05
TSHI						
Low	1			1		
Medium	1.43	0.78, 2.61	0.24	0.90	0.27, 2.96	0.86
High	3.36	1.97, 5.75	<0.05	2.97	1.09, 8.13	<0.05

Note: FT4 is for free thyroxine, FT3 is for free triiodothyronine, TSH is for thyroid-stimulating hormone, TT4RI is for thyrotroph thyroxine resistance index, TSHI is for thyroid-stimulating hormone index, and TFQI is for thyroid feedback quantile-based index.

**Table 3 diagnostics-13-02013-t003:** Univariate prediction model of macrosomia.

Index	AUC (95% CI)	Specificity	Sensitivity	Cut-Off
FT4	0.59 (0.53, 0.65)	0.89	0.34	19.15
FT3	0.68 (0.62, 0.73)	0.77	0.49	4.95
TSH	0.56 (0.50, 0.63)	0.91	0.28	2.87
FT3/FT4	0.58 (0.52, 0.63)	0.45	0.69	0.27
TFQI	0.64 (0.57, 0.69)	0.95	0.31	0.73
TT4RI	0.59 (0.52, 0.64)	0.67	0.52	30.70
TSHI	0.66 (0.62, 0.68)	0.98	0.19	3.57

Note: FT4 is for free thyroxine, FT3 is for free triiodothyronine, TSH is for thyroid-stimulating hormone, TT4RI is for thyrotroph thyroxine resistance index, TSHI is for thyroid-stimulating hormone index, and TFQI is for thyroid feedback quantile-based index.

## Data Availability

The data used to support the findings of this study are available from the corresponding author upon request.

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
