# Peer review of "Relationship between the Central and Peripheral Thyroid Sensitivity Indices and Fetal Macrosomia: A Cohort Study of Euthyroid Pregnant Women in China"

_diagnostics, 2023, doi:10.3390/diagnostics13122013_

Round 1

Reviewer 1 Report

Introduction: Authors should develop in this part a paragraph on the risk of fetal macrosomia, especially in GDM pregnancies; along with ref. 4 please consider an important review PMID: 27727018. One of the main concerns in pregnancies with macrosomia is that it has been demonstrated that macrosomic fetuses show different cardiovascular indices, in particular the Umbilical Artery Pulsatility Index, compared with non-macrosomic fetuses, therefore it is more difficult to manage these pregnancies and find the right timing for delivery (PMID: 26699801).

Lines 112-117: I would delete this part since the OGTT method is quite standardized in all the centers.

Results: Tables are usually considered a summary that does not replace the information in the text. please include in the manuscript not only the p value but also the statistical comparison, eg for BMI: (22.05±3.28 vs 22.14±2.91, c2=-1.95, p=0.05) so that readers can fully understand the results. In particular for BMI in the text authors stated that p is <0.05, but in the Table p=0.05, clarify this point.

Line 140: clarify is this is “maternal” (pregestational?) BMI

Line 169: are these adjusted OR? In this case you should refer to aOR instead of OR. Correct in the Table 2 Adjust with Adjusted.

Line 260: In recents studies it has been demonstrated that fetuses from women with pregestational diabetes or that will develop GDM later in pregnancy show an increased Fetal Heart Rate (FHR) in the early pregnancy (PMID: 30465928, PMID: 31706055). In the light of your results, can Authors postulate a role of TH in the origin of these difference in modulation of fetal heart?

Reviewer 2 Report

In this work, the authors sought to investigate the relationship between central and peripheral thyroid sensitivity indices and macrosomia in euthyroid pregnant women. The findings showed that both central and peripheral thyroid sensitivity in early pregnancy were independent risk factors. A prediction model was established for macrosomia using relevant indicators, which could help in preventing and treating macrosomia in clinical practice. These results are interesting as they provide a theoretical basis for further research and treatment of this condition. I would recommend the publication of this work as is. 

English quality is good and the sentences are easy to understand.  

Author Response

Dear Professor,

Thank you very much for your decision letter and advice on our manuscript entitled “Relationship between the Central and Peripheral Thyroid Sensitivity Indices and Fetal Macrosomia: A cohort study of euthyroid pregnant women in China”. We hope that the revision is acceptable for the publication in your journal.

Look forward to hearing from you soon.

With best wishes,

Yours sincerely,

Xin Zhao

Xiaomei Zhang